# Characterization of a Syngeneic Orthotopic Model of Cholangiocarcinoma by [^18^F]FDG-PET/MRI

**DOI:** 10.3390/cancers16142591

**Published:** 2024-07-19

**Authors:** Lena Zachhuber, Thomas Filip, Behrang Mozayani, Mathilde Löbsch, Stefan Scheiner, Petra Vician, Johann Stanek, Marcus Hacker, Thomas H. Helbich, Thomas Wanek, Walter Berger, Claudia Kuntner

**Affiliations:** 1Preclinical Imaging Lab (PIL), Department of Biomedical Imaging and Image-Guided Therapy, Medical University of Vienna, 1090 Vienna, Austria; lena.zachhuber@meduniwien.ac.at (L.Z.); thomas.wanek@meduniwien.ac.at (T.W.); 2Division of Nuclear Medicine, Department of Biomedical Imaging and Image-Guided Therapy, Medical University of Vienna, 1090 Vienna, Austria; 3Institute of Animal Breeding and Genetics & Biomodels Austria, University of Veterinary Medicine, 1210 Vienna, Austria; thomas.filip@vetmeduni.ac.at; 4Department of Pathology, Medical University of Vienna, 1090 Vienna, Austria; behrang.mozayani@meduniwien.ac.at; 5Core Facility Laboratory Animal Breeding and Husbandry, Medical University of Vienna, 1090 Vienna, Austria; 6Centre for Cancer Research and Comprehensive Cancer Center, Division of Applied and Experimental Oncology, Medical University of Vienna, 1090 Vienna, Austriawalter.berger@meduniwien.ac.at (W.B.); 7Medical Imaging Cluster (MIC), Medical University of Vienna, 1090 Vienna, Austria; 8Division of General and Pediatric Radiology, Department of Biomedical Imaging and Image-Guided Therapy, Medical University of Vienna, 1090 Vienna, Austria

**Keywords:** cholangiocarcinoma, orthotopic, syngeneic, mouse model, positron emission tomography (PET), magnetic resonance imaging (MRI), 2-deoxy-2-[^18^F]fluoro-D-glucose ([^18^F]FDG)

## Abstract

**Simple Summary:**

Cholangiocarcinoma (CCA) is a type of liver cancer with few treatment options and low survival rates in advanced stages. Our study developed a mouse model to study this cancer type by implanting CCA cells into the liver of mice. We used advanced imaging techniques (MRI and PET scans) to monitor tumor growth and metabolism over four weeks. We observed that tumors became visible early and grew steadily over time. PET scans showed increasing tumor activity, and blood tests revealed liver damage. Most mice developed lung metastases after four weeks. Our research shows that combining MRI and PET scans effectively tracks CCA progression in mice, providing valuable insights into cancer development and investigating potential treatments.

**Abstract:**

Cholangiocarcinoma (CCA) is a type of primary liver cancer originating from the biliary tract epithelium, characterized by limited treatment options for advanced cases and low survival rates. This study aimed to establish an orthotopic mouse model for CCA and monitor tumor growth using PET/MR imaging. Murine CCA cells were implanted into the liver lobe of male C57BL/6J mice. The imaging groups included contrast-enhanced (CE) MR, CE-MR with static [^18^F]FDG-PET, and dynamic [^18^F]FDG-PET. Tumor volume and FDG uptake were measured weekly over four weeks. Early tumor formation was visible in CE-MR images, with a gradual increase in volume over time. Dynamic FDG-PET revealed an increase in the metabolic glucose rate (MRGlu) over time. Blood analysis showed pathological changes in liver-related parameters. Lung metastases were observed in nearly all animals after four weeks. The study concludes that PET-MR imaging effectively monitors tumor progression in the CCA mouse model, providing insights into CCA development and potential treatment strategies.

## 1. Introduction

Cholangiocarcinoma (CCA) stands as the most prevalent biliary malignancy and the second most common primary hepatic malignancy, following hepatocellular carcinoma. These tumors represent a diverse set of aggressive malignancies originating from different locations of the biliary duct [1]. They are categorized according to their anatomical location as intrahepatic cholangiocarcinoma (iCCA) and extrahepatic cholangiocarcinoma, including perihilar (pCCA) and distal cholangiocarcinoma (dCCA). ICCAs are mass-forming and located proximal to the second-order bile ducts within the hepatic parenchyma (10–20%). Perihilar CCA emerges between the second-order bile ducts and the cystic duct, comprising the majority, approximately 50–60%, of CCAs across different studies. Distal CCA (dCCA) arises distal to the cystic duct insertion (20–30%) [2,3]. Each subtype has its unique epidemiology, molecular pathogenesis, and management strategy. Despite its relative rarity, constituting only 2–3% of all gastrointestinal cancers [4], the incidence of CCA is increasing in several countries globally, and given its high lethality with 5-year overall survival (OS) ranging from 7% to 20%, this disease has aroused considerable scientific interest [5]. Curative options, such as surgical resection (for iCCA) and liver transplantation, are viable only for patients diagnosed at early stages. For those with advanced disease not suitable for surgical intervention, the first-line treatment involves systemic chemotherapy employing gemcitabine and cisplatin. Recently, combination therapies such as gemcitabine and cisplatin with durvalumab or pembrolizumab have shown improvements in progression-free and overall survival (TOPAZ-1 [6] and KEYNOTE-966 trial [7]). In addition, second-line treatments of two different targetable mutations are available and approved, namely fibroblast growth factor receptor 2 (FGFR2) fusions and isocitrate dehydrogenase 1 (IDH1) [5]. However, there is still an urgent need for the development of targeted molecular therapies tailored to CCA. Such precision medicine approaches hinge on a deeper understanding of the molecular underpinnings of CCA.

In cancer research, in vitro studies utilizing cell culture are commonly employed to explore cancer cells’ genetic and cellular complexity. Besides monolayer cell cultures (2D cell culture), there is much research on developing organoid cultures (3D cell culture). In the last decades, the use of organoid cultures has significantly increased as they proved to be a powerful and reproducible tool for studying organogenesis, pathobiology, and drug development. While 2D mono-cultures offer advantages like high reproducibility, homogeneity, and tightly controlled experimental conditions, they do not accurately mimic the characteristic features of biliary tumors, like cell-to-cell and cell-to-matrix interactions, phenotypic heterogeneity, and the effect of the tumor microenvironment (TME) on cancer progression [8]. However, organoid cultures, more closely resembling in vivo physiology than monolayer cultures, do not entirely encompass the diverse biological processes taking place in tumors in vivo [9,10,11].

Consequently, studying interactions between different cell types within a tumor or investigating the role of various biological processes becomes challenging. Furthermore, exploring novel therapeutic targets necessitates preclinical studies in animal models. Thus, the utilization of in vivo models becomes imperative in cancer research. 

The choice of animal model for cholangiocarcinoma (CCA) should be guided by the question to be addressed and should ideally be reproducible in independent approaches. The ideal animal model of CCA would originate from the biliary tract in an immunocompetent host, possessing a microenvironment matched to the species, would be time-efficient, and would faithfully recapitulate the genetic, anatomical, and phenotypic features observed in human CCA [12]. 

For CCA, a variety of mouse and rat models have been established [8,12,13]. They are based on chemotoxic induction, genetically engineered models (GEMMs), or the implantation of human (xenograft) or mouse (allograft) tumor cells or tissue into the liver (orthotopic) or subcutaneously (heterotopic). Syngeneic models have the advantage of implanting rodent CCA cells into an animal of the same species, displaying a fully functional immune system. For human CCA, it was recently shown that circulating immune cells play an important role in the prognosis and chemotherapy response of patients with CCA [14]. Therefore, the syngeneic orthotopic model of CCA is of interest for the development of new therapeutic approaches. 

Orthotopic tumor models adequately reproduce the tumor microenvironment, typically exhibit faster early-stage tumor growth, and also include spontaneous metastases, reflecting their contribution to cancer aggressiveness. Furthermore, orthotopic models better reproduce human pharmacodynamics of drug responsiveness depending on the tumor location [8]. Therefore, orthotopic models are better predictors of clinical therapeutic outcomes [15]. A weakness of the orthotopic liver cancer model is the higher time-consuming development and difficulty in monitoring tumor progression [16]. However, modern imaging technologies such as positron emission tomography (PET) combined with magnetic resonance imaging (MRI) can be applied to follow tumor growth and tumor glucose metabolism for several weeks after tumor cell implantation. Therefore, the current study aimed to establish a relationship between tumor growth and tumor glucose metabolism in a syngeneic orthotopic CCA model. Therefore, we performed dynamic 2-deoxy-2-[^18^F]fluoro-D-glucose ([^18^F]FDG) PET imaging and hybrid [^18^F]FDG-PET/MR at weekly intervals in the generated mouse model. In addition, spontaneous metastases were evaluated on the extracted organs using hematoxylin and eosin staining.

## 2. Materials and Methods

### 2.1. Animal Model

The animal study was approved by the Austrian Federal Ministry of Education, Science and Research (project number 2021–0.611.621). Study procedures were conducted in accordance with the European Community’s Council Directive of 22 September, 2010 (2010/63/EU), and data reported in this study comply with the ARRIVE (Animal Research: Reporting of In Vivo Experiments) guidelines 2.0 [17]. 

Male C57BL/6J mice (*n* = 44) aged 8–10 weeks were obtained from Core Facility-Laboratory Animal Breeding and Husbandry (2325 Himberg, Austria) (*n* = 20; 22.6 ± 0.93 g) and Janvier Labs (*n* = 24; 27.2 ± 1.23 g). Animals were housed under controlled environmental conditions (21 ± 2 °C, 40–70% humidity, 12 h light/dark cycle) with different nesting and enrichment materials (red polycarbonate houses, aspen wood wool, nestlets, aspen wood sticks), free access to standard laboratory rodent diet (LASQdiet™ Rod16; Altromin Spezialfutter GmbH & Co. KG, Lage, Germany), and water. Animals were observed visually daily to evaluate their overall health condition and weighed once a week. An acclimatization period of at least one week was provided before the animals were used for the experiments. A detailed list of all animal procedures is given in Appendix A.

### 2.2. Cell Culture

The murine CCA cell line (SB-1) was kindly provided by the lab of Gregory Gores (Mayo Clinic, Rochester, NY, USA). SB-1 tumor cells express specific CCA markers as SRY (Sex Determining Region Y)-Box 9 (SOX9) and cytokeratin (CK)-7 and 19, but lack hepatocyte nuclear factor 4 alpha and alpha-smooth muscle actin, markers of hepatocellular carcinoma and cancer-associated fibroblasts [18]. They genetically resemble iCCAs found in a subset of patients [19]. The tumor cells were cultivated in Dulbecco’s Modified Eagle Medium (DMEM) (Gibco™ DMEM, high glucose, GlutaMAX™ Supplement, pyruvate; Fisher Scientific (Austria) GmbH), supplemented with 10% fetal bovine serum (FBS). They were kept in standardized conditions at 37°C in a humidified incubator in an atmosphere containing 5% CO_2_ during cultivation. As the original cell line was contaminated with mycoplasma, they were treated for two treatment cycles with a combination of pleuromutilin derivate and tetracycline derivative (BM-Cyclin, Roche, Cat. No. 10 799 050 001, Sigma-Aldrich, Merck KGaA, Germany) according to the internal standard operation protocol. After that, the tumor cells underwent regular mycoplasma testing through PCR analysis. To confirm that the treated cells were in accordance with the parental cell line, the IMPACT™ PCR profile and CellCheck^TM^ 19 tests were performed by an external lab (IDEXX BioAnalytics, IDEXX GmbH, Germany). SB-1 cells were checked daily (except on weekends) and passaged every 2–3 days during cultivation.

### 2.3. Establishment of the Orthotopic Syngeneic Mouse Model

SB-1 cells were cultivated for approximately 16 days and cells were not passaged for at least one day prior to inoculation. On the day of surgical inoculation, they were processed and prepared according to our standard operating protocol in serum-free DMEM for injection. They were stored on ice for a maximum of 2 h during inoculation [20]. 

Male C57BL/6J mice (*n* = 40) were anesthetized using an induction box filled with isoflurane (3–5%) in air. Once a sufficient depth of anesthesia was reached, the animal received an injection subcutaneously of Carprofen (Rimadyl^®^ ad us. vet. solution for injection, 50 mg/mL) 10 mg/kg or Buprenorphine (Temgesic^®^ injection solution, 0.3 mg/mL) 0.6 mg/kg, afterward the animal was shaved, positioned, and fixed on a heated plate. Anaesthesia was maintained via a head mask with integrated suction. The maintenance dose of isoflurane ranged between 1–2% by volume, depending on the required depth of anesthesia, based on respiration rate measurement. Under sterile conditions and deep anesthesia, a 1 cm incision was made below the xiphoid process to access the abdominal cavity. The superomedial aspect of the left medial liver lobe was visualized. Using a 30-gauge needle, 30 μL of serum-free DMEM containing 1 × 10^5^ tumor cells was slowly injected into the subcapsular region of the liver parenchyma in the liver lobe. To prevent leakage of the tumor cells and blood loss, a cotton swab was held over the injection site for one minute. Afterward, the abdominal wall and skin were closed in separate layers with absorbable suture material (Monosyn^®^ 5/0, DS 12; product number: C0022210, B. Braun Austria Gesellschaft m.b.H). Furthermore, after surgery, the mice received a subcutaneous injection of 0.5 mL isotonic electrolyte solution (Ringer-Lactat-Solution by Hartmann, B. Braun Austria Gesellschaft m.b.H). For postoperative analgesia, mice received Piritramid (Piritramid^®^ injection solution 7.5 mg/mL, Hameln Pharma GmbH, Germany) over drinking water for 3 days. We added 5% glucose (Glucose-Solution 5% ad us. vet. B. Braun Austria Gesellschaft m.b.H) to make the water more tasty. Animals were weighed daily for 4–5 days after surgery and clinically observed using our in-house scoring sheet for pain assessment. 

### 2.4. Experimental Design

The experimental design is depicted in Figure 1. Briefly, mice were randomly assigned to the four different study groups: contrast-enhanced (CE) MR, 60-min dynamic [^18^F]FDG-PET, and sequential static PET-MR using two different [^18^F]FDG administration routes (intraperitoneal injection—i.p. and intravenous injection—i.v.). An overview of the mice groups included in the study is given in Table 1. Imaging started 7.2 ± 1.0 days after the tumor cell inoculation and was continued weekly for 4 weeks. For the imaging procedures, mice were weighed and anesthetized in an induction chamber using isoflurane (4–5% in air). An overview of the animal weights over the study period is given in Appendix A. After that, we transferred the mice to a temperature-controlled double imaging chamber for PET imaging and a small rodent volume coil for MR imaging. An intravenous catheter was placed into the lateral tail vein for injection of the contrast agent or radiotracer. During preparation and measurement, animals were warmed, and anesthesia was maintained with an anesthetic facemask (isoflurane 1.5–2.5% in air) while respiration rate was monitored (SA Instruments Inc., Stony Brook, NY, USA). For all the PET imaging groups, the blood glucose level was measured before [^18^F]FDG injection and after the completion of the PET scan using a conventional glucometer (FreeStyle FREEDOM Lite, mg/dl, Abbott GmbH, Wiesbaden, Germany). The experimental methods used in this study adhere to published guidelines [21]. 

Following the last scan, a large blood sample was obtained by puncturing the retrobulbar plexus under anesthesia. The obtained blood was centrifugated (1500 rpm, 5 min) and plasma was stored at −18 °C until the analysis. After blood sampling, mice were sacrificed by cervical dislocation still under anesthesia, and the liver, including the tumor, was extracted. In addition, suspicious organs (e.g., lung, kidney, pancreas) with possible metastasis were extracted and used for histopathological assessment.

#### 2.4.1. Contrast-Enhanced MR Imaging

In the CE MR imaging group, 100 μL (diluted; 0.1 mL Primovist mixed with 0.9 mL sodiumchloride; 0.1 mmol/kg) of contrast-agent (CA) (Primovist^®^ 0.25 mmol/mL solution for injection, Bayer Vital GmbH, 51368 Leverkusen) were injected intravenously. Thereafter, anatomical images were obtained using a 1 Tesla Bruker ICON™ (Bruker Corporation, Ettlingen, Germany) scanner, a dedicated small animal system operating on ParaVision 6.1. Images of coronal sections were acquired using a T1-weighted flash sequence with a flip angle of 50°, 30 ms repetition time, and a 7 ms echo time. The field of view used was 76 × 28 × 24 mm with a 217 × 80 × 34 matrix, resulting in a voxel size of 0.35 × 0.35 × 0.7 mm^3^. The scan time was 7:36 min. The axial sections were acquired using a T1-weighted flash sequence with a flip angle of 50°, 32 ms repetition time, and a 7 ms echo time. The field of view used was 30 × 28 × 40 mm with an 86 × 80 × 53 matrix, yielding a voxel size of 0.35 × 0.35 × 0.755 mm^3^. The scan time was 12:54 min. A thin-slice thickness for the coronal section of 0.7 mm and for the axial section of 0.75 mm was used to provide detailed anatomical structure.

#### 2.4.2. PET Imaging

PET imaging was performed on a dedicated preclinical PET scanner (Focus 220^TM^, Siemens Healthineers, Knoxville, TN, USA) with 7.6 cm axial and 19 cm transaxial field-of-view. Two mice were imaged side by side in one PET image acquisition using a dual-mouse bed (m2m imaging Corp, Cleveland, OH, USA). Mice underwent a 60-min dynamic [^18^F]FDG scan using the injected activities in a volume of 100 μL, listed in Table 1. PET data acquisition was initiated at the start of intravenous injection (0.1 mL as slow bolus over ~40 s), and list-mode data were acquired for 60 min with an energy window of 250–750 keV and a 6 ns timing window. A 10-min transmission scan using a rotating ^57^Co-point source was performed before each PET scan to obtain data for attenuation correction.

#### 2.4.3. PET/MR Imaging

The PET-MR imaging group was divided into two groups; one group was given an intravenous CA injection (100 μL) followed by an intraperitoneal [^18^F]FDG injection (100 μL). The second group received 20 μL CA and 80 μL of [^18^F]FDG intravenously. The injected activities for both PET/MR groups are indicated in Table 1. After that, a coronal and axial MR image was acquired. Then, the imaging chamber (including the MR coil with the mouse) was transferred to the PET scanner, and a static PET image was acquired for 15 min, starting 45 min post-injection using the acquisition parameters given before.

### 2.5. Image Analysis

#### 2.5.1. MR Image Analysis

For image analysis, anatomic MR images were oriented in the standard orientation (head first, prone) and displayed in horizontal and axial directions. Then, the tumor was manually outlined on consecutive planes on the horizontal images using the software program AMIDE (amide.exe 1.0.4 [22]). Afterward, the position and size were controlled on the axial planes and corrected if necessary, and the tumor volume was recorded.

#### 2.5.2. Dynamic PET Image Analysis

Dynamic PET list-mode data from the 60-min scans were sorted into three-dimensional sinograms according to the following frame sequence: 8 × 5 s, 2 × 10 s, 2 × 30 s, 3 × 60 s, 2 × 150 s, 2 × 300 s, and 4 × 600 s. All PET images were reconstructed by Fourier rebinning of the 3D sinograms followed by two-dimensional ordered subset expectation maximization (OSEM) using 16 subsets and 4 iterations, resulting in a voxel size of 0.4 × 0.4 × 0.8 mm^3^. The standard data correction protocol, including normalization, attenuation, and decay correction, was applied to the data. For image analysis, images were corrected by the injected activity and body weight and expressed as standardized uptake value (SUV). Thereafter, organs of interest (tumor, brain, heart, liver, kidneys, vena cava) were defined by delineating manual or pre-defined volumes of interest (VOIs) using the software program AMIDE [22]. Organ sizes are given in Appendix A. Then, the time-activity curves (TACs) of these VOIs were extracted, and the areas-under-the-curves (AUCs) from 0 to 60 min were calculated. 

The curve derived from the vena cava ROI was used as an image-derived input function (IDIF, [23]). For that, the vena cava curve was scaled to the liver curve using the obtained values from the last time frame [24,25]. Afterwards, a time-dependent plasma-to-blood equilibrium ratio was applied to obtain the plasma input function [26]. The final obtained plasma input function $C_{p}\left(T\right)$ was used for calculating the net influx rate (*K_i_*) for the tumor and different organs derived from the slope of the linearized Patlak graphical analysis [27,28]: (1)$$\frac{C_{ROI}\left(T\right)}{C_{p}\left(T\right)}=K_{i}\times \frac{\int_{0}^{T}C_{p}\left(t\right)dt}{C_{p}\left(T\right)}+Int$$
where the *AUC_ROI_* from 0-T was used as a measure of $\int_{0}^{T}C_{ROI}\left(t\right)dt.$ The plot became linear after 10 min for all assessed organs and the tumor. From the obtained net influx rate, the metabolic rate of glucose (*MRGlu*) was calculated: (2)$$MRGlu=\frac{C_{glu}}{LC}\times K_{i}$$
where the $C_{glu}$ was the average blood glucose level of the two measurements at the beginning and the end of the scan given in mmol/L, and LC is the lumped constant. For the present study, the following values were used: LC = 1 (tumor), LC = 0.625 (brain, [29]), LC = 0.67 (heart, [30]). Patlak values were compared with semiquantitative uptake values given in *SUV* or ${SUV}_{glu}$, where ${{SUV}_{glu}=SUV\times C}_{glu}$.

#### 2.5.3. PET/MR Image Analysis

The static PET images were reconstructed, as mentioned before. Then, the PET images were coregistered to the MR images, and the tumor, lung, muscle, and brain were manually outlined on the MR images. VOI sizes for all analyzed organs are summarized in Appendix A. After that, the VOIs were copied to the PET image, and [^18^F]FDG uptake expressed as the mean (*SUV_mean_)* and maximum values (*SUV_max_*) of the VOI were extracted. Furthermore, the tumor volume was recorded. Finally, the *SUVglu* was calculated for the tumor (both for *SUV_mean_* and *SUV_max_*), taking into account the average blood glucose levels.

### 2.6. Histological Assessment

For histological assessment, the harvested organs were fixed in Histofix-4 solution (10:1) for 24 h and then transferred into 70% ethanol. Before processing, all samples were then transferred to buffered formalin (7.5%). The samples were processed and then embedded in paraffin (FFPE). The sections were cut at 2 µm thickness and manually stained according to standard Hematoxylin and Eosin (H&E) protocols. All sections were reviewed by a specialist gastrointestinal pathologist (B.M.). The liver parenchyma, which contained the primary CCA, was assessed, and all harvested organs were evaluated for potential metastatic deposits. These included pancreas, lung, kidney, and mesenteric fat. 

### 2.7. Blood Analysis

The plasma was stored at −18° until the blood chemistry analysis. Using a Cobas 4000 c311 analyzer for clinical chemistry (Roche Diagnostics, Mannheim, Germany), the following parameters were assessed: electrolyte panel: chloride (Cl), potassium (K), sodium (Na), phosphorus; liver function associated parameters such as alanine aminotransferase (ALT), aspartate aminotransferase (AST); alkaline phosphatase (ALP), bilirubin, albumin and total protein; kidney-function-associated parameters such as blood urea nitrogen (BUN) and creatinine (CREA); and triglycerides and glucose. 

### 2.8. Statistical Analysis

Statistical testing was performed using GraphPad Prism 10.2.2 software (GraphPad Software, La Jolla, CA, USA). Differences in organ uptake from the static [^18^F]FDG-PET/MR scans between the four weeks were analyzed by ordinary one-way ANOVA followed by Tukey’s multiple comparisons test. Blood chemistry parameters were analyzed by a 2-sided unpaired t-test with Welch correction using the Holm–Sidak method and assuming individual variance for each group. Correlation analysis was performed between outcome parameters of the kinetic modeling and glucose levels. The level of statistical significance was set to *p* < 0.05. Unless stated otherwise, all values are given as mean ± standard deviation (SD).

## 3. Results

### 3.1. Cell Culture

The treated SB-1 cells were shown to be free of *Corynebacterium bovis*, *Corynebacterium* sp. (HAC2), *Ectromelia virus* (ECTV), EDIM, *Hantaan virus*, K virus, LCMV, LDEV, MAV1, MAV2, mCMV, MHV, MNV, mouse kidney parvovirus (MKPV), MPV, MTV, MVM, *Mycoplasma pulmonis*, *Mycoplasma* sp., polyoma, PVM, REO3, sendai, and TMEV. No interspecies contamination was found. As no genetic profile was available for this specific cell line, the treated cells were compared with the parental cell line. The results from the cell authentication using short tandem repeat (STR) profiling are listed in Appendix B in Table A1.

### 3.2. Tumor Growth Derived by CE-MR 

The first CCA tumors were visible on the CE-MR image at around 7 days after surgery. CE-MR images from an exemplary mouse are shown in Figure 2A over the duration of the imaging experiments. 

### 3.3. PET Kinetic Modeling

VOI-based analysis of the dynamic small animal PET data was performed to determine tumor, brain, heart, and liver uptake of [^18^F]FDG. PET images over the four weeks of observation after tumor cell implantation are shown in Figure 2B. Organ-derived TACs are given in Appendix A. For the determination of the net influx (*K_i_*) as an outcome parameter for [^18^F]FDG distribution, we generated an image-derived blood input function (IDIF) by placing a cylindrical VOI over the vena cava of the individual animals. Then, this curve was scaled to the liver curve using the obtained values from the last time frame and transferred to the plasma input function by multiplication of a time-dependent plasma-to-blood equilibrium ratio. An exemplary patlak plot from a tumor is shown in Appendix A. The relationships between blood glucose level and *K_i_*, *MRGlu*, and *SUV* in the tumor, heart, and brain are illustrated in Appendix A. There was a significant inverse relationship between tumor *K_i_* (R^2^ = 0.2807, *p* = 0.0054), heart *K_i_* (R^2^ = 0.1818, *p* = 0.0133), and cerebral *K_i_* (R^2^ = 0.1856, *p* = 0.0123), and blood glucose level. However, no significant association was found between the tumoral *MRGlu* and cerebral *MRGlu* and blood glucose level, as shown by the quality of regression lines (R^2^ = 0.00026 and R^2^ = 0.01209, respectively), and the regression slopes were not significantly different from zero. On the contrary, for the heart *MRGlu*, a significant positive correlation with blood glucose level was obtained (R^2^ = 0.1696, *p* = 0.0173). *SUV* did not correlate significantly with blood glucose levels for all analyzed regions.

For the tumor, both *SUV* and *SUVglu* exhibited a significant positive correlation with MRGlu, as shown in Figure 3 (*SUV*: R^2^ = 0.2478, *p* = 0.0113; *SUVglu*: R^2^ = 0.2123, *p* = 0.0205). 

Finally, we obtained an increase in the *MRGlu* tumor values over the observation period of four weeks, as illustrated in Appendix A. All these results together suggest that both *SUV* and *SUVglu* are adequate representatives of *MRGlu* tumor values for the static [^18^F]FDG-PET/MR scans.

### 3.4. Tumor Volume and FDG Uptake from the PET/MR Groups

The obtained tumor volumes from all the CE-MR images (MR group and the two [^18^F]FDG-PET/MR groups) are displayed in Figure 4. The tumor volume gradually increased over the observation period with a doubling rate of around 6.4 days.

Apart from the tumor volume, we also extracted the [^18^F]FDG uptake in the tumor after i.p. or i.v. [^18^F]FDG injection. Exemplary images from the i.p. [^18^F]FDG-PET/MR and i.v. [^18^F]FDG-PET/MR groups can be found in Figure 2C,D. In the i.p. group, we had some misinjections into the gut. These images were excluded from the analysis. From the remaining images, the *SUV* and *SUVglu* were calculated for the tumor (both for SUV_mean_ and SUV_max_), taking into account the average blood glucose levels. The results for the i.p. [^18^F]FDG-PET/MR and i.v. [^18^F]FDG-PET/MR groups can be found in Figure 5 for each individual animal. For both groups, an increase in the *SUV_max_* and *SUVglu_max_* was observed, whereas the *SUV_mean_* and *SUVglu_mean_* increased until week 3, and exhibited a small decline in week 4. This is probably due to necrotic and/or hypoxic areas in the tumor, reducing the average [^18^F]FDG SUV in the total tumor region. 

In addition to the tumor, we also analyzed changes in the [^18^F]FDG uptake in other organs, such as the brain, lung, and muscle, for the i.v. [^18^F]FDG-PET/MR group. As depicted in Figure 6, we obtained statistically significant higher [^18^F]FDG uptake given in *SUV* in week 4 compared to week 3 and week 1 for the brain. Moreover, lung [^18^F]FDG uptake in week 4 drastically increased compared to the rest of the imaging time points. No changes in muscle uptake were observed. When the *SUV* parameter was corrected by the average blood glucose level yielding *SUVglu*, only one statistically significant difference remained for the lung.

Furthermore, we also observed some experimentally induced and pathological changes visualized in the [^18^F]FDG images. In the first week after surgery, a higher [^18^F]FDG activity was found at the site of the wound suture, pointing to physiological inflammation. Moreover, as shown in Figure 2D, in some animals, the [^18^F]FDG uptake after i.p. injection was very high in the abdominal cavity at four weeks after the surgery, whereas the uptake in the rest of the body, such as in the heart and brain, were low. The autopsy revealed ascites in the abdominal cavity in these animals, which seems to have reduced the uptake of [^18^F]FDG into the bloodstream and thus distribution to all the other organs. 

### 3.5. Blood Chemistry and Histopathology

Blood chemistry performed on the plasma samples taken on the last imaging day revealed a change in the liver-associated parameters, as shown in Figure 7. ALP, AST, and ALT parameters were statistically significantly higher in tumor-bearing mice compared to control (healthy) mice of the same strain, sex, and age, whereas albumin was lower in the CCA model compared to the control. In addition, glucose values measured from the plasma sample correlated with the blood glucose value directly taken after the scan (Appendix A).

For the characterization of the orthotopic CCA mouse model, we sacrificed the animals (*n* = 18/24, Janvier lab animals) after 4 weeks of imaging and harvested the liver, which included the tumor and other organs suspected of containing metastatic deposits. The selected samples were analyzed by a specialist GI histopathologist based on H&E-stained slides. CCA tumor cells were confirmed in all analyzed liver samples (*n* = 17/17). The CCA in the mice livers was comparable to the small duct iCCA of the mass-forming type in humans, contrasting with the periductal infiltrating large duct type that typically grows in the perihilar region. The tumors were morphologically identical between mice and humans, exhibiting small tubular, solid, and occasionally poorly differentiated growth patterns. Interestingly, within the primary tumor, metaplastic bone (*n* = 16/17) and, in one instance, metaplastic cartilage formation (*n* = 1/17) was noted (see Figure 8). In addition to the liver, tumor metastases were observed in nearly all analyzed lung samples (*n* = 16/17). Most of the metastases were multifocal, and some contained necrotic areas. Metastatic cells were also observed in the peritoneal cavity and in mesenteric fatty tissue in the vicinity of the kidney and pancreas. 

## 4. Discussion

Preclinical models of cholangiocarcinoma (CCA) are essential for accelerating the development of novel clinical treatment strategies. While ectopic xenograft or syngeneic mouse models—based on human or rodent cell lines injected into immunocompromised or immunocompetent mice—are easy to establish and have limited complications from the procedure, they have significant limitations. These cancer models typically reflect advanced tumor stages, grow rapidly, and make the study of early CCA challenging. Additionally, different CCA cell lines exhibit varying tumorigenic activity, with some unable to generate tumors after injection. Furthermore, these tumors are implanted in a non-physiological site, seldom metastasize, and may lose the molecular heterogeneity characteristic of human CCA [8]. We aimed to establish a syngeneic orthotopic mouse model of CCA to address these limitations. Recently, a CCA model using murine cells (SB1–7) derived from Akt-YAP driven tumors was described. These cells exhibit phenotypic features of human CCA, and their implantation results in the development of orthotopic tumors that are morphologically and phenotypically similar to human CCA [18]. This mouse model, having a fully functional immune system, is ideal for studying tumor-stroma interactions and testing immunotherapy-based interventions.

The motivation for this study was to examine the tumor growth behavior of orthotopic cholangiocarcinoma cell implantation (SB-1) in greater detail using in vivo preclinical imaging. Specifically, we aimed to investigate changes in tumor volume and glucose metabolism over four weeks following tumor cell implantation. To achieve this, we employed contrast-enhanced MR and [^18^F]FDG-PET imaging. Additionally, we sought to develop an imaging protocol suitable for subsequent therapeutic studies, ensuring the anesthesia duration was limited to a maximum of 1.5 h. Although this animal model was proposed six years ago, little is known about its tumor growth behavior, highlighting the importance of our research. Rizvi et al. [18] reported on tumor weight following orthotopic SB-1 cell (1 × 10^6^ cells, 40 µL, standard media) implantation into male C57BL/6 mice, measuring mean tumor weights of approximately 50 mg and 570 mg at two and four weeks post-implantation, respectively. However, their study included only two time points and did not incorporate in vivo imaging. Wabitsch et al. [19] established an orthotopic model by injecting SB-1 cells (2 × 10^5^ cells, 20 µL, 50:50 solution of PBS and Matrigel) into the left liver lobe of female 8-week-old C57BL/6 mice. They reported a tumor weight of approximately 800 mg in the control group 20 days post-implantation. In our study, CE-MR derived tumor volumes across all animals (*n* = 14–20 per time point) were approximately 18 mm^3^, 44 mm^3^, 127 mm^3^, and 332 mm^3^ at one-, two-, three-, and four-weeks post-implantation (1 × 10^5^ cells, 30 µL, DMEM), respectively. Given the variations in cell numbers, injection volumes, media, and implantation techniques and skills, these differences are acceptable and provide a basis for selecting the therapy start point in subsequent studies.

We utilized dynamic [^18^F]FDG-PET and Patlak graphical analysis to calculate quantitative outcome parameters, focusing particularly on the *K_i_* and *MRGlu* values. We observed that *K_i_* is strongly correlated with blood glucose for all analyzed organs, aligning with previous findings [31]. In contrast, tumor *MRGlu* did not correlate with blood glucose, and thus it was selected as the primary outcome value. This was an important validation, especially since we did not fast the animals before the [^18^F]FDG-PET scans, although numerous publications [26,32,33] recommend fasting periods before imaging. Fasting imposes a significant burden on the animals [34], and to minimize the loss of mice, we opted to measure blood glucose levels before and after the scans for glucose correction.

Given that dynamic [^18^F]FDG-PET combined with MRI on separate scanners requires prolonged anesthesia times (15 min for preparation, 60 min for the PET scan, 10 min for the attenuation scan, and 30 min for the MR scan), we aimed to use static [^18^F]FDG-PET scans. This approach allows MR scans to be performed during the uptake period. Consequently, we correlated the obtained *MRGlu* values with *SUV* and *SUVglu*, confirming a significant correlation. However, it should be noted that correcting for blood glucose levels results in higher data variability. This was illustrated in Figure 6, where the coefficients of variation (CV) were up to three times higher in the brain for *SUVglu* (CV ~ 16–31%) compared to *SUV* (CV ~ 7–9%). Similarly, in the lung region, the CV increased after correction for blood glucose, resulting in fewer statistically significant differences. Therefore, both *SUV* and *SUVglu* were selected as outcome parameters for the static [^18^F]FDG-PET scans. 

We also evaluated which administration route for [^18^F]FDG is better suited for this animal model. Previous studies have shown that both intravenous (i.v.) and intraperitoneal (i.p.) [^18^F]FDG injections result in similar tracer distribution approximately 30–60 min post-injection [26,32,35]. However, due to the risk of misinjections and pathological [^18^F]FDG uptake in the peritoneal cavity related to ascites, we ultimately opted for the i.v. injection route for the [^18^F]FDG-PET/MR protocol.

The most striking finding was the observation of metastasis in the lungs of nearly all analyzed animals. This result is not surprising, as orthotopic grafts are more likely to trigger tumor dissemination, leading to the development of distant metastases [36]. However, lung metastases had been reported previously but had not been published yet [8], underscoring the significance of our study. In orthotopic liver tumor models, lung metastasis appears to be common, as demonstrated in two immune-competent orthotopic hepatocellular carcinoma mouse models [37]. The lung metastases were macroscopically visible during the section and were confirmed by histopathological examination. Additionally, the enhanced [^18^F]FDG uptake in the lungs observed at 4 weeks post-implantation (see Figure 6b) further confirmed pathological uptake. Therefore, the combination of anatomical imaging (CE-MR) and molecular imaging ([^18^F]FDG-PET), alongside histopathology, proved to be an ideal method for characterizing this animal model. 

Furthermore, histopathological appraisal identified metaplastic bone and cartilage formation in the primary CCA tumors, a finding exceptionally rare in human CCA. Many primary tumors and lung metastasis showed areas of tumor necrosis. Peritoneal dissemination was also frequently observed, as demonstrated in the spread surrounding the pancreas and kidneys.

Finally, ex vivo analysis revealed increased ALT, AST, and ALP levels in tumor-bearing mice. When hepatocyte injury occurs, ALT is released from the damaged hepatocytes, causing a significant increase in serum ALT activity. Elevated ALT is associated with increased severity of liver diseases in humans [38], and similar increases have been reported in animal studies [37,39]. This biochemical evidence supports the presence of liver damage and further validates the use of this model for studying tumor growth and metastasis in the context of liver disease.

One limitation is that our study included only male mice and did not include an assessment of tumor growth and metabolism in female mice. We originally intended to include female mice in the study. However, the animal ethics committee did not approve our application to use female mice in accordance with good scientific practice at that time. The committee based its decision on the origin of the tumor cell line SB-1, which was generated in male mice, although this justification is no longer valid. We intended to conduct this study in female mice, mainly because of the documented differences between male and female subjects in preclinical and clinical studies [40,41], particularly in therapeutic investigations.

## 5. Conclusions

The combination of CE-MR and [^18^F]FDG-PET imaging proves effective for monitoring tumor growth and metabolism in the syngeneic orthotopic CCA tumor model. Notably, this model showcases lung metastasis formation observed four weeks post-tumor cell implantation. Additionally, liver tumors exhibit bone and cartilage formation, with elevated ALT, AST, and ALP levels confirming liver damage. With these findings, along with the established imaging protocols, this animal model is now primed for therapeutic studies, offering a promising avenue for exploring potential treatments.

## Figures and Tables

**Figure 1 cancers-16-02591-f001:**
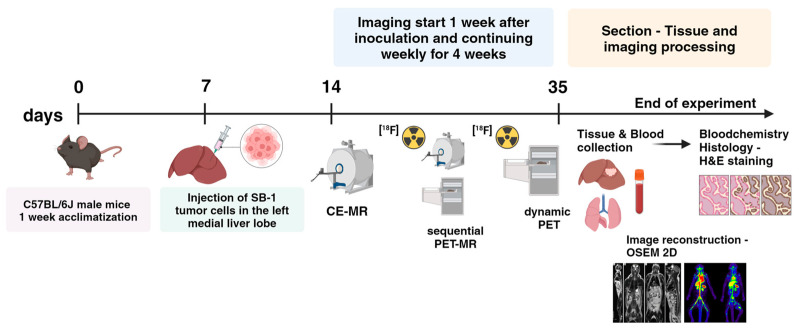
Graph illustrating the experimental design of the study. (Created with BioRender.com; www.biorender.com/ accessed on 18 June 2024).

**Figure 2 cancers-16-02591-f002:**
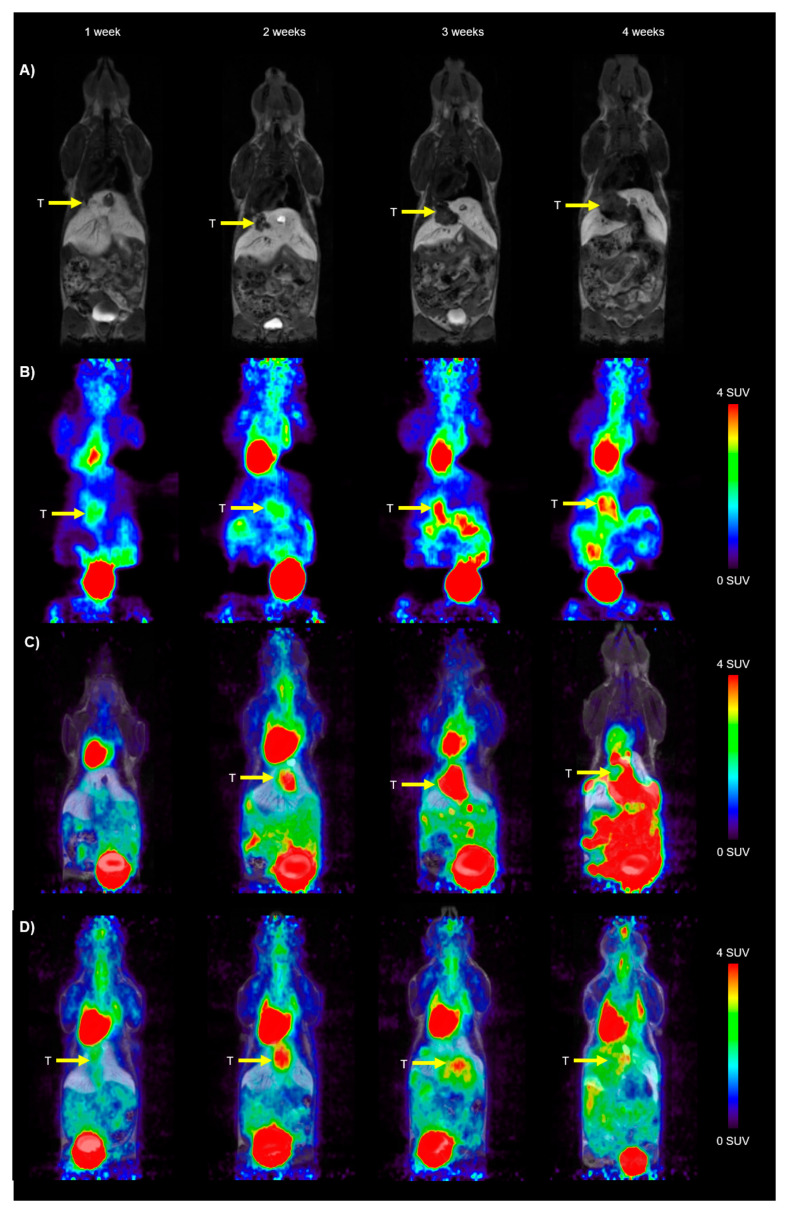
Representative horizontal images of the syngeneic orthotopic CCA model from the (**A**) contrast-enhanced (CE) MR group, (**B**) dynamic [^18^F]FDG-PET group (50–60 min time frame), (**C**) combined [^18^F]FDG-PET/MR group (i.p. injection), and (**D**) combined [^18^F]FDG-PET/MR group (i.v. injection). The radiation scale is set from 0–4 SUV. The tumor (T) is indicated with a yellow arrow on the images.

**Figure 3 cancers-16-02591-f003:**
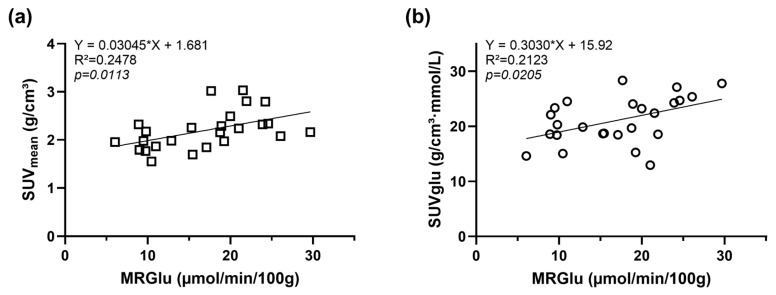
Correlation of MRGlu derived from the tumor by Patlak graphical analysis with (**a**) SUV and (**b**) SUVglu.

**Figure 4 cancers-16-02591-f004:**
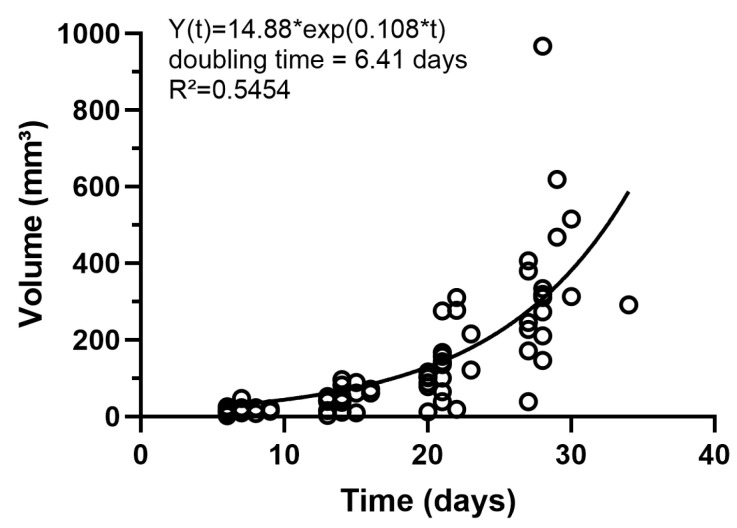
Tumor growth curve from the syngeneic orthotopic CCA model obtained by CE-MR imaging. Imaging was initiated around one week after tumor cell inoculation into the liver and performed weekly for four weeks.

**Figure 5 cancers-16-02591-f005:**
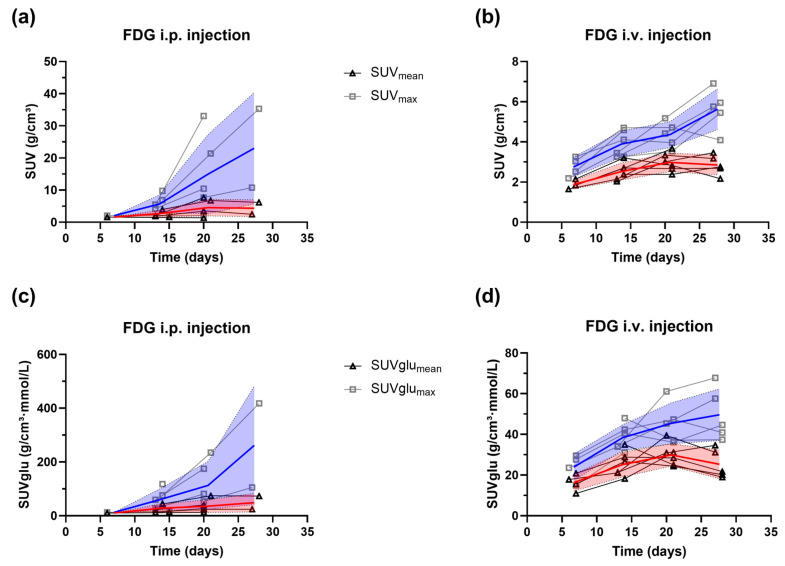
Tumor uptake given in *SUV* and *SUVglu* calculated from the *SUV_mean_* (triangles, red shades) and *SUV_max_* (squares, blue shades). The *SUVglu* values were corrected by the average blood glucose levels in the syngeneic orthotopic CCA model. The mean and the standard deviation (shaded area) are illustrated on the images for the (**a**,**c**) i.p. [^18^F]FDG-PET/MR group (*n* = 1–5) the (**b**,**d**) i.v. [^18^F]FDG-PET/MR group (*n* = 4–6). Bold lines indicate mean values. Imaging was initiated around one week after tumor cell inoculation into the liver and performed weekly for four weeks.

**Figure 6 cancers-16-02591-f006:**
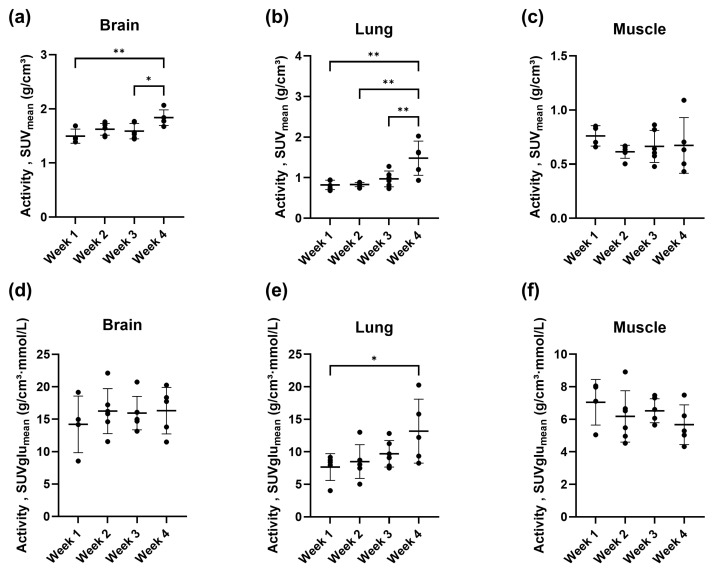
Activity concentrations given in SUV_mean_ for the (**a**) brain, (**b**) lung, and (**c**) muscle region derived from the i.v. [^18^F]FDG-PET/MR group (*n* = 5–6) in the syngeneic orthotopic CCA model. The SUV parameter corrected by the blood glucose levels, SUVglu, are shown for the (**d**) brain, (**e**) lung, and (**f**) muscle region. Imaging was initiated around one week after tumor cell inoculation into the liver and performed weekly for four weeks. The individual values and mean ± standard deviation are illustrated. * *p* < 0.05, ** *p* < 0.01, ordinary one-way ANOVA followed by Tukey’s multiple comparisons test.

**Figure 7 cancers-16-02591-f007:**
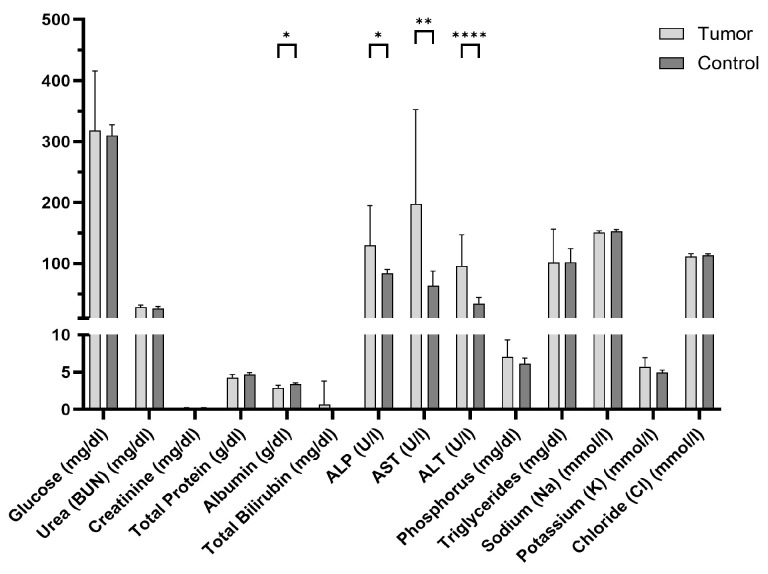
Blood chemistry parameter extracted from tumor-bearing mice four weeks after tumor cell implantation. Values are compared to parameters from control mice of the same strain, sex, and age. The bar plot shows the mean value and the standard deviation. * *p* < 0.05, ** *p* < 0.01, **** *p* < 0.0001; unpaired t-test with Welch correction.

**Figure 8 cancers-16-02591-f008:**
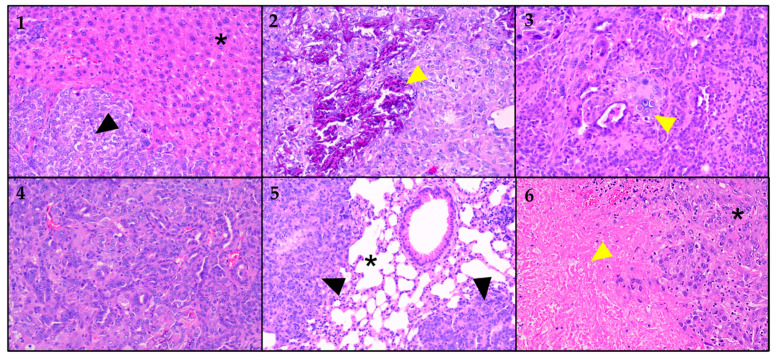
H&E-stained slices at different magnifications; images of different organ samples. (**1**) The star (*) denotes non-neoplastic liver parenchyma. The black arrow lies within an area of cholangiocarcinoma (CCA). (**2**) Liver parenchyma with cholangiocarcinoma (CCA). The yellow arrow points to an area of metaplastic bone formation. (**3**) Liver parenchyma with cholangiocarcinoma (CCA). The yellow arrow points to an area of metaplastic cartilage formation. (**4**) Higher magnification of liver with cholangiocarcinoma (CCA). (**5**) Lung parenchyma (*) with CCA metastases, marked by black arrows. (**6**) Liver with cholangiocarcinoma (*). The yellow arrow denotes the necrotic area.

**Table 1 cancers-16-02591-t001:** Overview of group size (*n*), weight, activities, and blood glucose levels for the [^18^F]FDG imaging experiments given for each experimental week. The PET scans from the two PET/MR groups were started 45 min post injection.

Imaging Weeks	1	2	3	4
	MR			
*n*	6	6	6	6
Weight (g)	26.2 ± 2.5	27.0 ± 2.9	27.2 ± 3.2	27.2 ± 3.2
MR scan (days after surgery)	7.7 ± 1.0	15.0 ± 0.9	22.0 ± 0.9	29.0 ± 0.9
	PET dynamic			
*n*	10	9	9	7
Activity at PET scan (MBq)	9.0 ± 5.8	9.6 ± 1.8	10.3 ± 2.6	11.0 ± 3.1
Weight (g)	25.9 ± 3.0	26.9 ± 2.6	27.0 ± 2.7	26.9 ± 3.6
Blood glucose (mmol/L)	9.9 ± 1.7	10.0 ± 2.2	10.0 ± 3.4	10.7 ± 2.3
PET scan (days after surgery)	7.2 ± 1.0	14.1 ± 1.1	21.1 ± 1.1	27.9 ± 1.1
	PET/MR (i.p.)			
*n*	6	7	7	6
Activity at PET scan (MBq)	4.8 ± 0.8	7.8 ± 2.9	6.6 ± 1.7	5.7 ± 3.0
Weight (g)	26.3 ± 2.8	26.3 ± 2.4	26.1 ± 2.6	26.3 ± 3.6
Blood glucose (mmol/L)	7.5 ± 1.2	9.3 ± 2.0	8.4 ± 2.0	9.7 ± 1.8
PET/MR scan (days after surgery)	7.0 ± 1.1	13.6 ± 0.8	20.6 ± 0.8	27.3 ± 0.5
	PET/MR (i.v.)			
*n*	6	6	6	5
Activity at PET scan (MBq)	7.6 ± 1.3	7.7 ± 0.9	7.8 ± 1.4	8.6 ± 1.0
Weight (g)	28.7 ± 0.7	29.3 ± 1.0	29.1 ± 1.4	29.7 ± 1.6
Blood glucose (mmol/L)	9.0 ± 2.0	10.1 ± 2.6	10.0 ± 1.2	8.8 ± 1.5
PET/MR scan (days after surgery)	6.7 ± 0.5	13.7 ± 0.5	20.7 ± 0.5	27.6 ± 0.5

## Data Availability

The raw data supporting the conclusions of this article will be made available by the authors on request.

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
