# Peer review of "Characterization of a Syngeneic Orthotopic Model of Cholangiocarcinoma by [18F]FDG-PET/MRI"

_cancers, 2024, doi:10.3390/cancers16142591_

Round 1
Reviewer 1 Report
Comments and Suggestions for Authors
1. What is the meaning of i.p. 18F-FDG PET shown in the articles?
2. Why do the liver tumors exhibit bone and cartilage formation?
3. The glucose level of several mice seems to exceed the standard values, close to the DM level. Will it impact the SUVs of tumors and organs and physiological activity?
4. What does the dynamic PET study mean in the experiment design?
5. The necrosis caused decreased SUV of tumors. Is it significant? Can it be possible for other factors?
Reviewer 2 Report
Comments and Suggestions for Authors
This is an interesting study to present an orthotopic mouse model of cholangiocarcinoma. The mouse cell line was from Ref 20, which is one of the seven cell lines derived from the YAP-driven murine CCA model. The Introduction did not clearly state why SB-1, not the other 6 cell lines, was chozen, and exactly what clinical model this one represents: iCCA, pCCA or dCCA, and at what stage? To be really useful, especially for testing treatment strategies, the paper needs to present what type/stage of CCA this mouse model is modeling? Why not use YAP-driven murine CCA models directly for both male and female mice (SB-1 was derived from male?!)?
Regarding histology, how were compared to the corresponding human CCA? Please discuss.
Round 2
Reviewer 1 Report
Comments and Suggestions for Authors
None
Reviewer 2 Report
Comments and Suggestions for Authors
The authors addressed the points raised by tis reviewer.